# Discovery of neutralizing SARS-CoV-2 antibodies enriched in a unique antigen specific B cell cluster

Stine Sofie Frank Lende[1,2]*, Nanna Møller Barnkob[3], Randi Westh Hansen[3], Harsh Bansia[4], Mike Vestergaard[3], Frederik Holm Rothemejer[1,2], Anne Worsaae[3], Deijona Brown[4], Maria Lange Pedersen[1,2], Anna Halling Folkmar Rahimic[2], Anna Karina Juhl[1], Torben Gjetting[4,5], Lars Østergaard[1,2], Amédée Des Georges[4,6,7], Laurent-Michel Vuillard[8], Mariane Høgsbjerg Schleimann[1], Klaus Koefoed[3], Martin Tolstrup[1,2]*

1 Department of Infectious Diseases, Aarhus University Hospital, Skejby, Denmark, 2 Department of Clinical Medicine, Aarhus University, Aarhus, Denmark, 3 Symphogen, Ballerup, Denmark, 4 Structural Biology Initiative, CUNY Advanced Science Research Center, New York, NY, United States of America, 5 Antibody Technology, Novo Nordisk A/S, Måløv, Denmark, 6 Department of Chemistry and Biochemistry, City College of New York, New York, NY, United States of America, 7 PhD Programs in Biochemistry, and Chemistry, Graduate Center, City University of New York, New York, NY, United States of America, 8 Servier, Suresnes, France

* stsoni@rm.dk (SSFL); mtol@clin.au.dk (MT)

**Data Availability Statement:** All relevant data are within the paper and its Supporting Information files.

## Abstract

Despite development of effective SARS-CoV-2 vaccines, a sub-group of vaccine non-responders depends on therapeutic antibodies or small-molecule drugs in cases of severe disease. However, perpetual viral evolution has required continuous efficacy monitoring as well as exploration of new therapeutic antibodies, to circumvent resistance mutations arising in the viral population. We performed SARS-CoV-2-specific B cell sorting and subsequent single-cell sequencing on material from 15 SARS-CoV-2 convalescent participants. Through screening of 455 monoclonal antibodies for SARS-CoV-2 variant binding and virus neutralization, we identified a cluster of activated B cells highly enriched for SARS-CoV-2 neutralizing antibodies. Epitope binning and Cryo-EM structure analysis identified the majority of neutralizing antibodies having epitopes overlapping with the ACE2 receptor binding motif (class 1 binders). Extensive functional antibody characterization identified two potent neutralizing antibodies, one retaining SARS-CoV-1 neutralizing capability, while both bind major common variants of concern and display prophylactic efficacy *in vivo*. The transcriptomic signature of activated B cells harboring broadly binding neutralizing antibodies with therapeutic potential identified here, may be a guide in future efforts of rapid therapeutic antibody discovery.

## Introduction

Throughout the SARS-CoV-2 pandemic, immense efforts have resulted in the rapid development and administration of vaccinations successfully reducing COVID-19 mortality [1].

**Funding:** This study was supported by a grant from the Danish Ministry for Research and Education (grant# 0238-00001B to MT) and The Danish Innovation Fund (grant# 0208-00018B to MT and KK). SSFL was supported by a scholarship from Aarhus University. The funders had no role in study design, data collection and analysis, decision to publish, or preparation of the manuscript.

**Competing interests:** The authors have declared that no competing interests exist.

However, several studies show the presence of a small group of primarily immunocompromised patients displaying vaccine non -or hypo-responses [2–7].These individuals, already at increased risk of severe disease from SARS-CoV-2, are consequently additionally vulnerable in the event of SARS-CoV-2 exposure and infection [8, 9]. This underlines the need for availability of effective treatment options and potential alternative prophylaxis.

Parallel to vaccine development, the pursuit of identifying SARS-CoV-2 monoclonal antibodies (mAbs) with therapeutic capacity have fast-tracked discovery and testing of such antibodies. The first to gain EUA and FDA approvals were the REGN-COV-2 dual mAbs combination Casirivimab and Imdevimab, showing reduced risk of Covid-19 related hospitalization and death [10]. Since then, several mAb treatments have been approved and administered to COVID-19 patients, demonstrating both clinical safety and efficacy [11–14]. Common to both vaccines and mAb treatment is the targeting of the SARS-CoV-2 Spike surface protein. The Spike receptor-binding domain (RBD) in particular, due to its interaction with the human ACE2 receptor facilitating viral cell fusion and infection, has been shown to be a dominant immunogenic target [15–19].

However, the continued evolution of SARS-CoV-2 has proven to be a constant challenge with the emergence of new viral variants of concern (VOC), carrying mutations in the Spike protein. This has rendered otherwise clinically successful mAbs ineffective or severely reduced their viral neutralization potency [20]. Overall, the abrupt dynamics of the SARS-CoV-2 pandemic demands equally agile development of potent mAbs retaining neutralization breadth across new VOC. This requires discovery platforms facilitating rapid mAb discovery, as tools for enabling continued adaptation to the ongoing pandemic and future emerging pathogens.

In this study, we analyzed and cloned B cells from 15 SARS-CoV-2 convalescent participants using LIBRA-seq technology (linking B cell receptor to antigen specificity through sequencing), profiling single B cells with respect to antigen-specificity by sequencing oligo barcodes [21]. We identified a transcriptomic cluster of activated memory B cells highly enriched for cells producing SARS-CoV-2 neutralizing antibodies. Following screening and characterization of mAbs originating from SARS-CoV-2 recovered participants, we report the discovery of two broadly neutralizing mAbs with *in vivo* prophylactic efficacy.

## Results

### Participant enrollment and selection

To obtain cell material for B cell receptor characterization and mAb cloning, a cohort of 194 SARS-CoV-2 recovered participants infected during the spring of 2020 was considered, for which the clinical and immunological characteristics have been previously described [17, 22]. From this data, PBMC's from 15 participants with high plasma neutralization capacity were selected for B cell isolation and antibody discovery (Fig 1). An overview of the 15 participants' demographics and clinical characteristics is listed in Table 1. Of the included participants, 40% were female, with a group median age of 52 years (range 31–67). Participants experienced COVID-19 symptoms for a median of 16 days (range 0–47), and more than half (53.3%) of the selected group had been hospitalized during their disease course. Participants were included at a minimum of 14 days after full recovery with a median time from diagnosis to PBMC sampling of 40 days (range 24–67).

### Transcriptomic profiling of antigen-specific B cells

To identify antigen (Ag)-specific B cells, we used FACS to isolate B cells bound to four different coronavirus Ag conjugated dextramers: SARS-CoV-2 RBD, trimer and D614G mutated trimer, and SARS-CoV-1 RBD. The sorted B cells were profiled with respect to gene expression

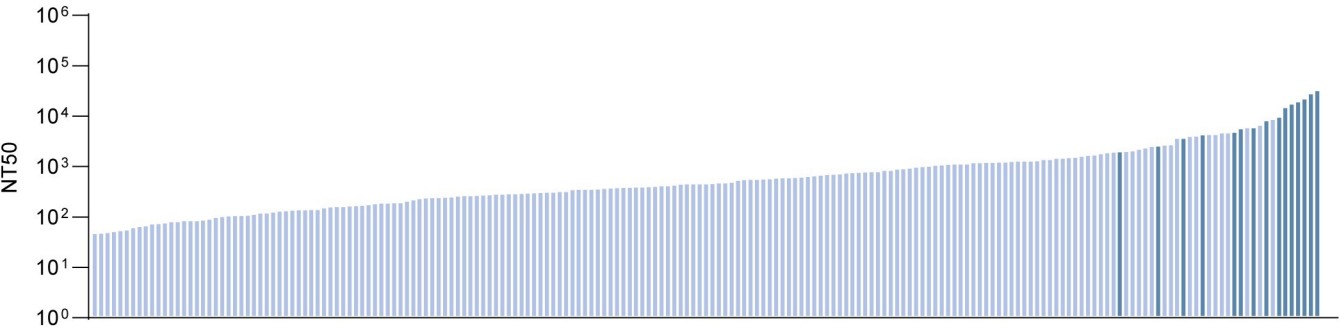

**Fig 1. Cohort selection and clinical characteristics.** 50% Neutralization titer ($NT_{50}$) values (plasma dilution factor) from 194 SARS-CoV-2 recovered participants calculated from SARS-CoV-2 pseudovirus neutralization curves are shown on y-axis. Participants are ordered along the x-axis from lowest (left) to highest (right) $NT_{50}$ values within the cohort. The 15 participants chosen for B cell receptor characterization are highlighted.

(GEX), variable antibody sequence (V(D)J) and barcode-based Ag score using single-cell RNA sequencing. After quality control, a total of 7176 cells were visualized in two dimensions using UMAP, based on their GEX count matrix. Of these cells 61,6% were from group 1, 9,1% were from group 2, and the remaining 29.3% were from group 3. Unsupervised clustering further revealed nine transcriptomic clusters (Fig 2A). Manual annotation based on marker genes revealed six clusters of memory B cells (clusters 1, 2, 3, 4, 5 and 9) accounting for 74.6% of the total population, two clusters of activated memory B cells (6 and 8) accounting for 18.2%, and one cluster of naïve B cells (cluster 7), accounting 7.1%. The most prevalent isotype was IgA1 (44.5%), followed by IgG1 (19.4%) and IgA2 (18.9%). These isotypes were all associated with high somatic hyper-mutation (SHM) rate when compared to the inferred naïve germline sequence (Fig 2B). In contrast, the few IgD (0.3% cells) and IgM B cells (11.1% cells) were associated with a low mutation rate and found predominantly in the naïve cluster 7 (Fig 2C). The *IGHV* germline usage per isotype group is summarized in S3A Fig, with *IGHV3-23* and *IGHV3-33* being the most frequently utilized IGHV-germline genes in the B cell population concordant with previous observations [23]. Curiously, otherwise reported *IGHV3-66* and *IGHV3-53* that are associated with class I neutralizing mAbs most commonly seen in convalescent individuals [19, 24–26], were used in only 1.74% and 1.17% of the isolated cells, respectively [27–29].

**Table 1. Demographics and clinical characteristics of included participants.**

| Demographics and Clinical Characteristics of included participants | | |
|---|---|---|
| **Characteristics** | | **n = 15** |
| Age, years, median (range) | 52 | (31–67) |
| Female sex, no (%) | 6 | (40) |
| COVID-19 disease severity, no (%) | | |
| 1. Home/outpatient, no limitation of daily activities (asymptomatic/mild) | 2 | (13.3) |
| 2. Home/outpatient, limitation of daily activities (moderate) | 5 | (33.3) |
| 3. Hospitalized (severe) | 8 | (53.3) |
| Duration of COVID-19 symptoms, days, median (range) | 16 | (0–47) |
| Time from diagnosis to inclusion, day, median (range) | 40 | (24–67) |

Table displaying basic demographics and clinical characteristics of the 15 included participants. COVID-19 disease severity groups were defined as follows: 1) asymptomatic individuals, experiencing no limitations in daily activities. 2) Moderately sick participants with disease limiting daily activities, able to recover at home. 3) Hospitalized participants, irrespective of ICU admission and/or oxygen supplementation.

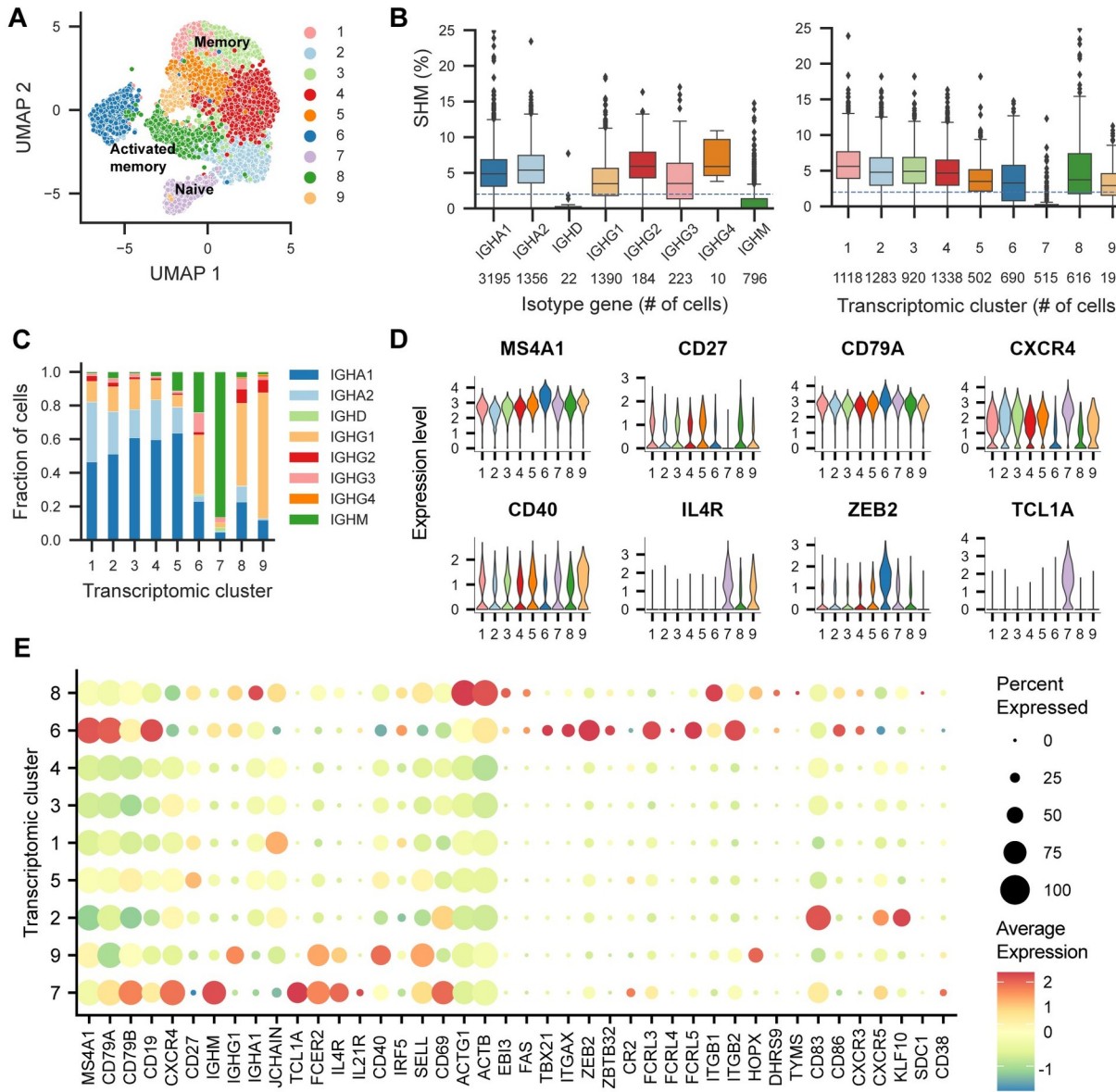

**Fig 2. Single-cell profiling of 7176 B cells from three donor groups by scRNA-seq and V(D)J-seq.** (A) UMAP projection and unsupervised clustering revealing 9 transcriptomic clusters annotated according to selected marker genes. (B) Somatic hyper-mutation percentage compared to inferred naïve germline stratified by isotype and transcriptomic cluster, highlighting one cluster of naïve B cells associated with particularly low mutation rate. The dotted line indicates the threshold used for the selection of mAbs for validation. The number of cells within each cluster or isotype is displayed below. (C) Fraction of cells with each isotype stratified by transcriptomic cluster highlighting the naïve cluster and three IgG-rich clusters. (D) mRNA expression levels given by the log of the normalized UMI counts of selected markers in each transcriptomic cluster. (E) Scaled average expression in each cluster indicating memory B cell, naïve B cell and activated B cell markers highlighting unique transcriptomic profiles of the clusters. The size of the dots indicates the percentage of cells expressing the given marker gene within the cluster.

All clusters showed a wide expression of the B cell markers *CD79A*, *CD79B*, *CD19* and *MS4A1* (CD20) (Fig 2D and 2E). The naïve cluster 7 was defined by expression of the markers *TCL1A*, *FCER2*, *IGHM*, *IL21R* and *IL4R* and increased expression of *CXCR4*, which orchestrates entrance into lymphoid organs [30]. Furthermore, the naïve cluster was associated with low to zero expression of *CD27*, otherwise widely expressed in the remaining memory B cell population (Fig 2E). Cluster 9 also featured high expression of *IL4R* and *FCER2*, reduced

*CD27* expression, and upregulation of *CD40*, alongside the largest proportion of IgG antibodies in a cluster.

Clusters 6 and 8 comprising activated B cells showed expression of a range of activation markers previously reported [23, 31]. Both cluster 6 and 8 were significantly associated with expression of *CD86*, and *FAS* (encoding CD95), as well as the chemokine receptor CXCR3 associated with migration to inflamed tissue, and *EBI3* which is involved in regulating the germinal center reactions. Both clusters showed a higher fraction of IgG antibodies compared to the remaining clusters except cluster 9. Cluster 6 was defined by up regulation of transcription factors *ZEB2* and *ZBTB32* and the two integrin molecules $\alpha_X$ *ITGAX* (CD11c) and *ITGB2*. Furthermore, cluster 6 was associated with expression of *TBX21* encoding T-Bet, as well as *FCRL5* and *FCRL3*.

Compared to cluster 6, cluster 8 showed increased expression of the transcription factor *HOPX*, *ACTB*, activation markers *ACTG1* and the integrin beta 1 molecule (*ITGB1*). Of note, the transcription factor *KLF10* and activation marker *CD83* was significantly upregulated in the neighboring memory B cell cluster 2, highlighting the homogeneity in the transcriptomic profiles of the cell population. A minor fraction of cells expressed *SDC1* (CD138) and *CD38*, suggesting that a few plasma cells, despite having low levels of membrane-bound Ig, were also included in the Ag-specific sorting. These were not abundant enough to form a cluster.

Finally, correlation of the Ag scores with the transcriptomic clusters revealed a minor enrichment for the SARS-CoV-2 RBD Ag score in cluster 8 ($p$-value = 0.0133, Kruskal-Wallis non-parametric test) and a strong enrichment in cluster 6 ($p$-value = 5.117e-12). For the SARS-CoV-2 mutant D614G trimer Ag score, there was no measurable enrichment in cluster 8 ($p$-value = 0.818), but a strong enrichment in cluster 6 ($p$-value = 5.9252e-11) (S3B, S3C Fig). No significant differences were found between the scores for the SARS-CoV-1-associated Ags or the wild-type SARS-CoV-2 trimer.

## Monoclonal antibody selection and screening

We then set out to select mAbs from the B cell population that could be tested functionally. To focus the selection, three major criteria were defined that identified mAbs as eligible for screening: First, we considered only B cells of switched isotype, specifically IgG1, IgA1 and IgA2. Second, only mAbs mutated more than 2% from the germline sequence (as indicated in Fig 2B) were considered. Third, all antibodies with common drug development liability sequence motifs (defined in methods) were deselected. Utilizing the transcriptomic profiles, we completely deselected any B cells originating from the naïve cluster 7. It was hypothesized that the set of activated memory B cells could be responding to the cognate antigen and selected all eligible mAbs from cluster 6 and cluster 8. Finally, a broad selection of mAbs from the memory B cell populations was made. mAbs were selected from the full range of Ag scores. The selection of 455 mAbs that were produced as individual antibodies is summarized in S3D Fig.

Next, we screened all selected 455 mAbs for their functional neutralization of SARS-CoV-2 pseudovirus particles. Neutralization capacity of the mAb pool ranged from 100–0% neutralization of pseudovirus, as shown in Fig 3A. A cutoff of 80–95% neutralization was applied to define a group of "neutralizers" (containing nine mAbs), with a group of "top neutralizers" defined as virus neutralization above 95% (containing 24 mAbs). Interestingly, when mapping these selected mAbs back on the original B cell clustering, many of the neutralizing antibodies found, were sourced from cluster 8 (Fig 3B). This is also apparent when examining the number of virus neutralizing mAbs within each cluster division, with 1.73%, 18.52% and 5.75% of mAbs in clusters 6, 8 and all remaining clusters (excluding cluster 7) respectively, being

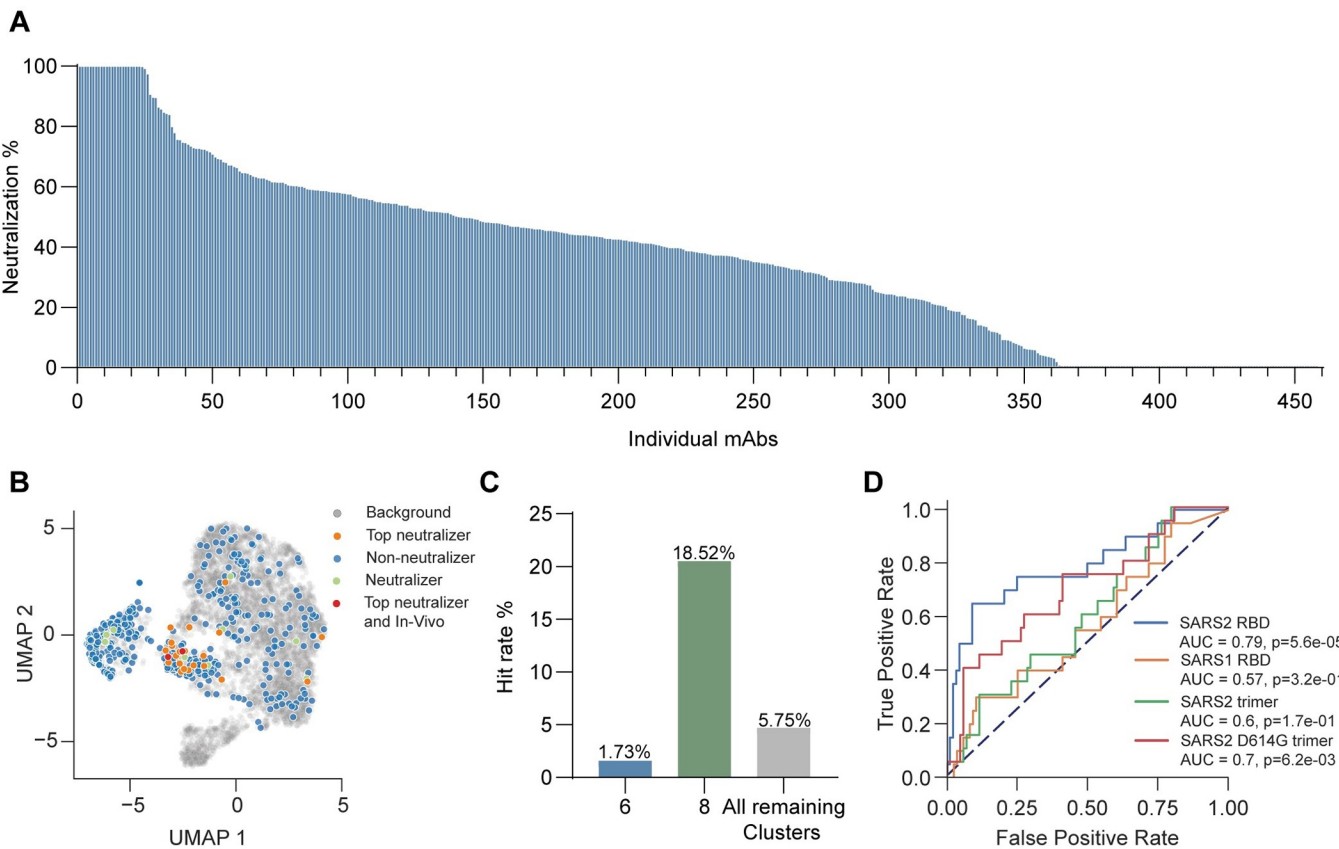

**Fig 3. Monoclonal antibody screening.** (A) Neutralization percentage of SARS-CoV-2 pseudovirus shown for each individual mAb supernatant analyzed at 20 µg/ml, shown on y-axis. MAbs are ordered along the x-axis from best (left) to poorest (right) neutralizers. n = 455. Screening was performed once in duplicate determinations. (B) Visualization of the expressed mAbs B cell cluster origin and distribution within all isolated B cells. Neutralizers shown in green (<80%-95%> neutralization, n = 9). Top neutralizers shown in red (<95% neutralization, n = 24). Non-neutralizers shown in blue (<80% neutralization). Background shown in grey (cells not expressed for screening).(C) Distribution of successful neutralizing mAbs, between clusters 6, 8 and remaining clusters (cluster 7 excluded). Hit rate cut-off for defining successful neutralizations was set at 80% pseudovirus neutralization. Hit rates were calculated within each cluster group. n = 455 (D) Predictive performance of Ag scores used for SARS-CoV-2 specific B cell sorting towards neutralization capability in cluster 8. n = 108. Red = SARS-CoV-2 D614G mutant trimer, Blue = SARS-CoV-2 RBD, Orange = SARS-CoV-1 RBD, Green = SARS-CoV-2 trimer. The p-value is based on a Kruskal-Wallis test of the receiver-operator characteristics curve.

successful neutralizing antibodies (Fig 3C). Although the Ag scores were not initially enriched in cluster 8 (S3B, S3C Fig), we evaluated their predictive performance using the area under the receiver operating characteristic curve (ROC-AUC) by applying a threshold to the Ag score and considering neutralization or binding as the positive class. We found that the Ag scores derived from both SARS-CoV-2 RBD and SARS-CoV-2 mutant D614G trimer correlated significantly with both neutralization (Fig 3D) and binding (S3E Fig). The Ag scores associated with SARS-CoV-1 RBD and SARS-CoV-2 trimer were not predictive. Antigen scores, somatic hypermutation and antibody isotype is listed for selected antibodies in S2 Table.

## In vitro analysis of lead mAbs

We further analyzed the neutralizing mAb pool to determine neutralization potency for each mAb, as well as their sensitivity to VOC emerging throughout the SARS-CoV-2 pandemic. An additional promising antibody, 29044, identified in a pilot run was added to the final pool of neutralizing mAb candidates. By performing full SARS-CoV-2 pseudovirus neutralization curves for each mAb, we determined their half-maximum inhibitory concentration ($IC_{50}$) and

ranked them from most (lowest IC$_{50}$) to least (highest IC$_{50}$) potent, as displayed in Fig 4A. Through mesoscale analysis, we determined mAb binding ability to SARS-CoV-2 Spike, RBD and N-Terminal Domain (NTD) (Fig 4A). The vast majority of neutralizing mAbs were RBD binders, with only a single mAb (31318) binding the NTD. Interestingly, three of the mAbs; 31259, 31319 and 31414, displayed SARS-CoV-1 Spike binding in addition to SARS-CoV-2 during SPR analysis (S1 Table). When examining the mAbs ability to retain binding towards common VOC RBD's using mesoscale analysis, these same antibodies retain binding sensitivity to both the Alpha, Beta, and Delta strains (Fig 4B). Broad binding to VOC was also observed for these mAbs during SPR analysis (S1 Table). In comparison, the majority of mAbs (58%) display a reduction of up to 10-fold in binding sensitivity during mesoscale analysis towards Beta and Gamma strains, compared to SARS-CoV-2 RBD binding.

Likewise, the ability of the most potent neutralizing mAbs to out-compete ACE2—Spike binding showed a greater loss of sensitivity to Beta and Gamma strains, compared to Alpha and Delta and also generally an even greater loss of ACE2 competition to Omicron BA.1 (Fig 4C). The ability to compete for ACE2 binding for all viral variants tested was observed for two mAbs; 29044 and 31259.

## Characterization of neutralizing mAbs epitopes

To evaluate the epitope coverage of the top neutralizing mAbs, we performed epitope binning by paired antibody competition analysis using SPR. To further determine the mAbs Spike binding regions, we included a panel of antibodies with structurally known epitopes; Sotrovimab [32], Imdevimab [33], Casivirimab [33], S2E12 [34] and S2H97 [34] (S6A Fig). Based on their competition profiles (S6B Fig), antibodies were clustered as six epitope communities with distinct competition patterns and visualized as a community plot (Fig 5A). These were further grouped into four main groups: Class 1 (Magenta), class 2 (Orange/yellow), class 3 (Green), and class 4 (Blue).

Overall, the majority of mAbs investigated in the binning assay compete for a highly similar epitope (S6A Fig). The vast majority of mAbs are found in an epitope community with class 1 binding mode and compete either with mAbs grouped in class 3 or mAbs in community class 2. Class 3 and the class 2 mAbs are non-overlapping, indicating that mAbs in each group bind distinct epitopes. The S2H97 reference mAb class 4 represents a unique epitope not targeted by any of the tested mAbs. Class 1 contains the reference mAbs Casivirimab and S2E12, both of which are known to bind the receptor-binding motif (RBM) of the RBD. Class 2 contains reference mAbs Sotrovimab and Imdevimab, which bind epitopes outside the RBM. Both class 2 and class 3 contained mAbs displaying no or weaker ACE2 competition compared to RBM binding mAbs (S6C Fig). The class 2 mAbs as well as the class 1 mAb (29044) were specifically distinguished by their conserved binding to the Omicron BA.1 VOC (Fig 5B). Additionally, all three mAbs in epitope community class 2 retained cross-binding to SARS-CoV-1 Spike RBD.

Finally, we report the Cryo-EM structure of SARS-CoV-2 spike trimer (B.1.1.529/Omicron) complexed with the Fab fragment of 29044 at overall resolution of 3.82 Å (Fig 5C). Overall topology of the Omicron BA.1 spike trimer in the complex reveals two up-RBDs and one down-RBD with each up-RBD having additional density corresponding to a molecule of Fab 29044. However, one of the bound Fabs lacks modellable densities presumably because of the conformational dynamics of Fab-up-RBD in that monomer of the Omicron BA.1 spike (Fig 5C) and hence was not used for modelling the variable region (Fv) of mAb 29044 (Fig 5D) Further, due to preferred orientation of the sample on the grid, density maps do not have enough definition to describe the antigen-antibody interface at the level of residue-residue interaction. PDBePISA [35] analysis of the interface in the Omicron BA.1 spike trimer complexed with

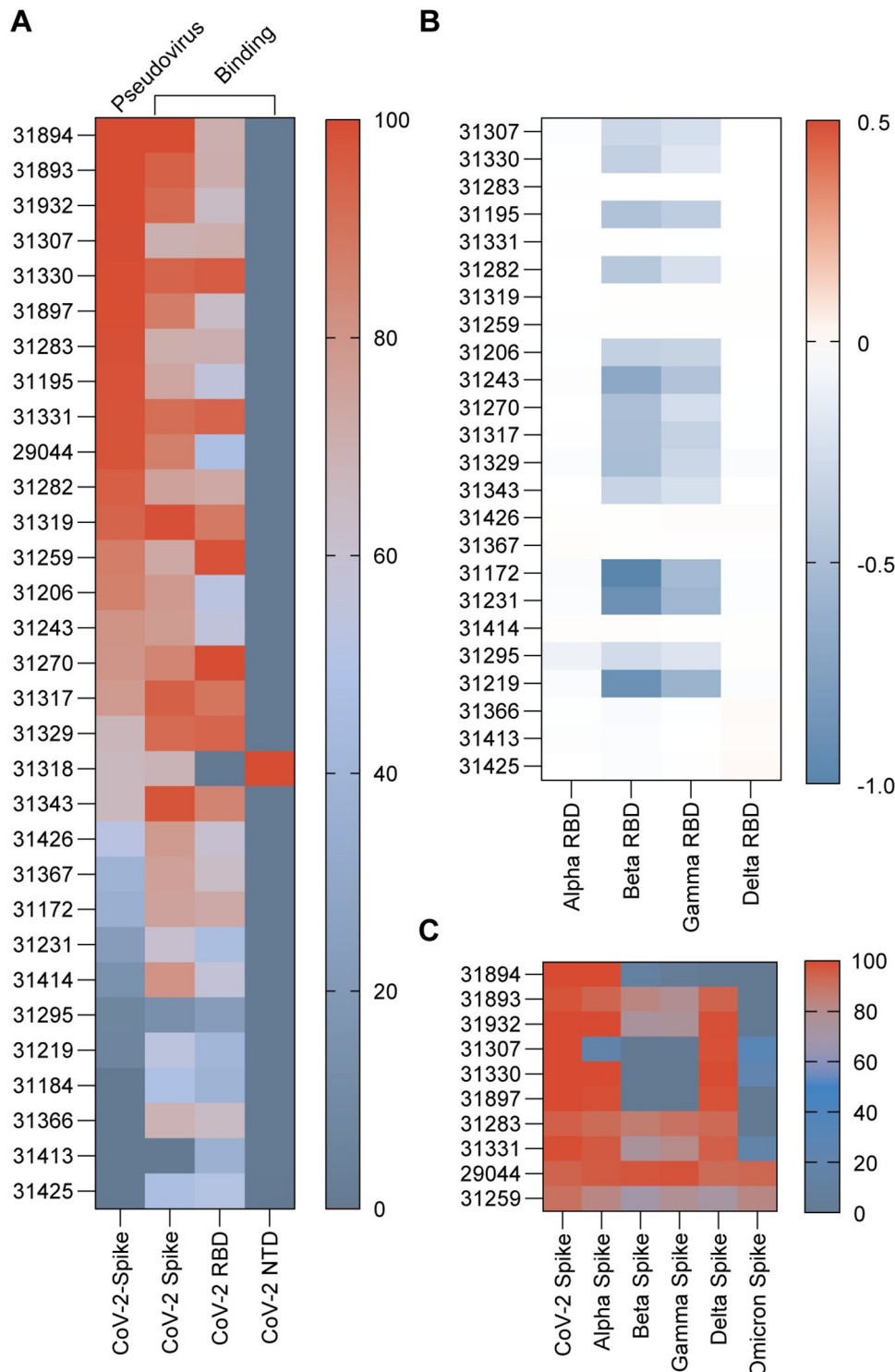

**Fig 4. In vitro analysis of lead mAbs binding.** (A) Ranking of purified mAbs from lowest $IC_{50}$ pseudovirus neutralization value at the top (best neutralization) to highest $IC_{50}$ at the bottom (poorest neutralization). Each antibody $IC_{50}$ value obtained from triplicate point determinations of the dilution curve. Mesoscale binding values are shown for each mAb (supernatant) towards SARS-CoV-2 Spike, N-terminal domain (NTD) and receptor binding domain (RBD), as a heat-map. The binding determinations were performed once in duplicate. Colors indicate normalization from 0–100 within each column. (B) Mesoscale binding values for binding to the RBD of viral variants Alpha (N501Y, A570D), Beta (K417N, E484K, N501Y), Gamma (K417T, E484K, N501Y) and Delta (L452R), shown as

fold change from SARS-CoV-2 RBD binding within each mAb individually. The binding determinations were performed once in duplicate. (C) Heat-map showing percentage ACE2 blocking for each mAb binding viral variant spike proteins (CoV-2, Alpha (B.1.1.7), Beta (B.1.351), Gamma (P.1), Delta (B.1.617.2) and Omicron BA.1(B.1.1.529). The ACE2 blocking analysis was performed in duplicate determinations of a dilution curve.

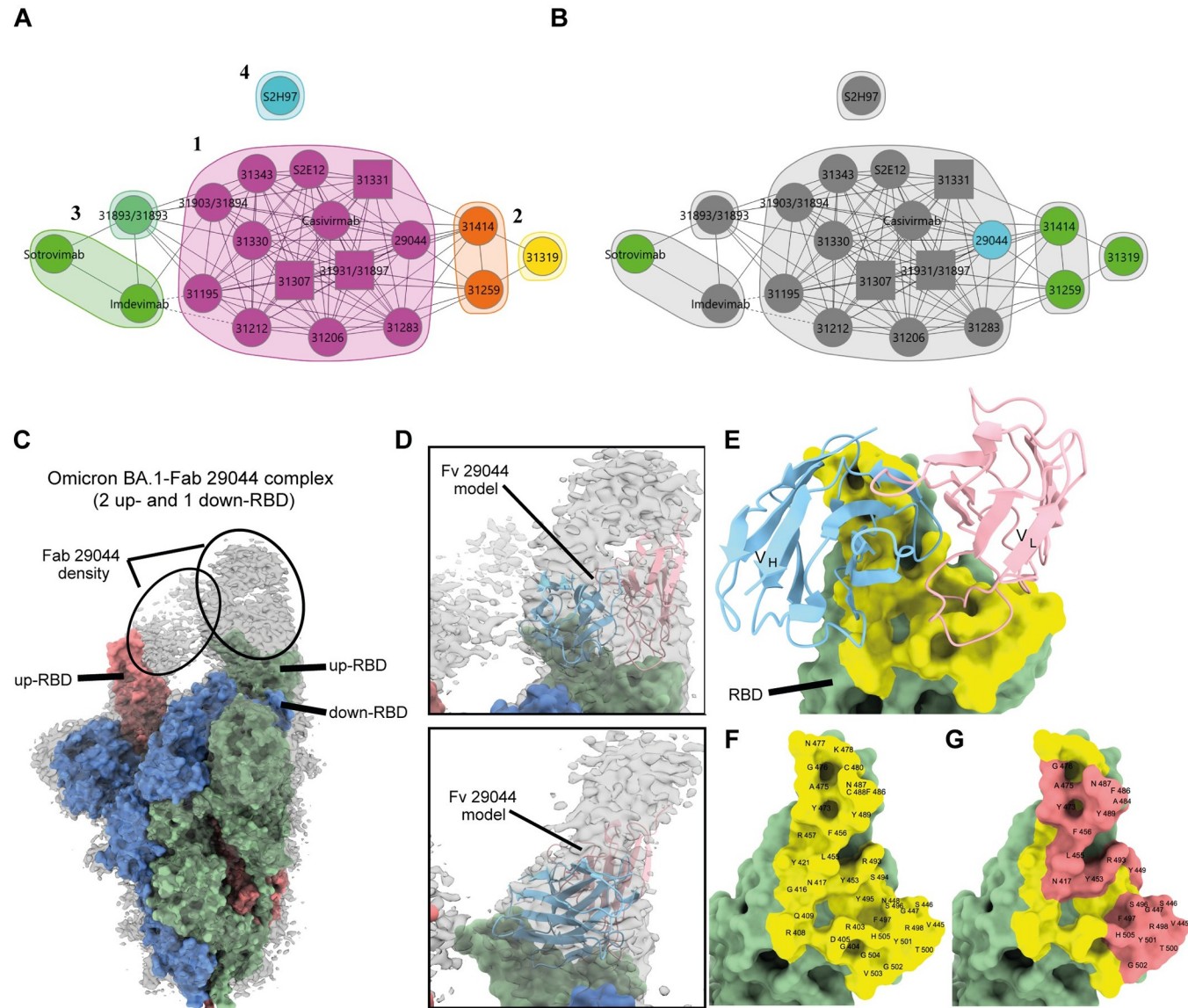

**Fig 5. Characterization of mAb epitopes and cryo-EM structure of Omicron BA.1 spike trimer with Fab 29044.** (A) Epitope network plot colored by communities. Antibodies are represented as nodes and by number (Circles are mAbs tested in both orientations and squares are mAbs tested only in one direction). Pairwise blocking relationships are indicated by chords, and dashed lines are mAbs showing asymmetric blocking. A cut-off of 3.5 (S6B Fig) was used to define epitope communities. Community class III (Greens) and class II (Oranges) compete with community class I (magenta) but not with each other. (B) Epitope community plot highlighting mAbs retaining binding to Omicron B.1.1.529 and SARS-CoV-1 RBD colored in green. MAbs retaining binding to Omicron B.1.1.529 RBD is shown in light blue. (C) Surface representation of spike trimer overlayed with the cryo-EM density map (grey) is shown. Additional density observed for the Fab 29044 is circled and up- (red and green monomer) and down-RBD (blue monomer) are highlighted. (D) Zoomed in views showing model corresponding to variable region (Fv) of mAb 29044 (cartoon representation; V$_H$, cyan; V$_L$, pink) fitted into the overlayed Cryo-EM map (grey). (E) Close-up view of the complex depicting binding interface (yellow). (F) Labelled RBD residues (yellow) interfacing with 29044 Fv are shown. (G) Labelled ACE2 footprint (coral) on RBD in the background of 29044 Fv interface is shown.

Fab 29044 reveals that the interface overlaps significantly with the ACE2 binding footprint (Fig 5E–5G).

## Neutralization capacity of elite mAbs

Two SARS-CoV-2 pseudovirus neutralizing antibodies; 29044 ($IC_{50}$ = 0.032ug/ml) and 31259 ($IC_{50}$ = 0.2253 ug/ml) blocked ACE2 binding to all viral variants tested, and were chosen as elite neutralizing mAb candidates, alongside the more potent, but non-Omicron cross-reactive mAb 31283 ($IC_{50}$ = 0.012 ug/ml). Further investigation revealed similar ACE2 –SARS-CoV-2 Spike inhibition titration curves for all three mAbs, while Sotrovimab (a non-ACE2 competitive mAb) showed poor inhibition in comparison (Fig 6A). Congruent with ACE2 –Spike competition screening data, full ACE2 –SARS-CoV-2 Omicron BA.1 Spike inhibition titration curves confirmed a loss of ACE2 competition for mAb 31283 (Fig 6B).

Verification of pseudovirus neutralization was performed with authentic SARS-CoV-2, where all three mAbs display similar $IC_{50}$ values ($31259_{IC50}$ = 0.54 ug/ml, $31283_{IC50}$ = 0.008 ug/ml, $29044_{IC50}$ = 0.073 ug/ml) compared to pseudovirus neutralization inhibition (Fig 6C). Variant sensitivities were functionally verified by pseudovirus neutralization for all three mAbs as well, (Fig 6D–6F) towards SARS-CoV-2 wild type, Alpha, Beta, Delta, and Omicron BA.1 and BA.2 strains as well as SARS-CoV-1. This confirmed the ability of mAb 31259 to

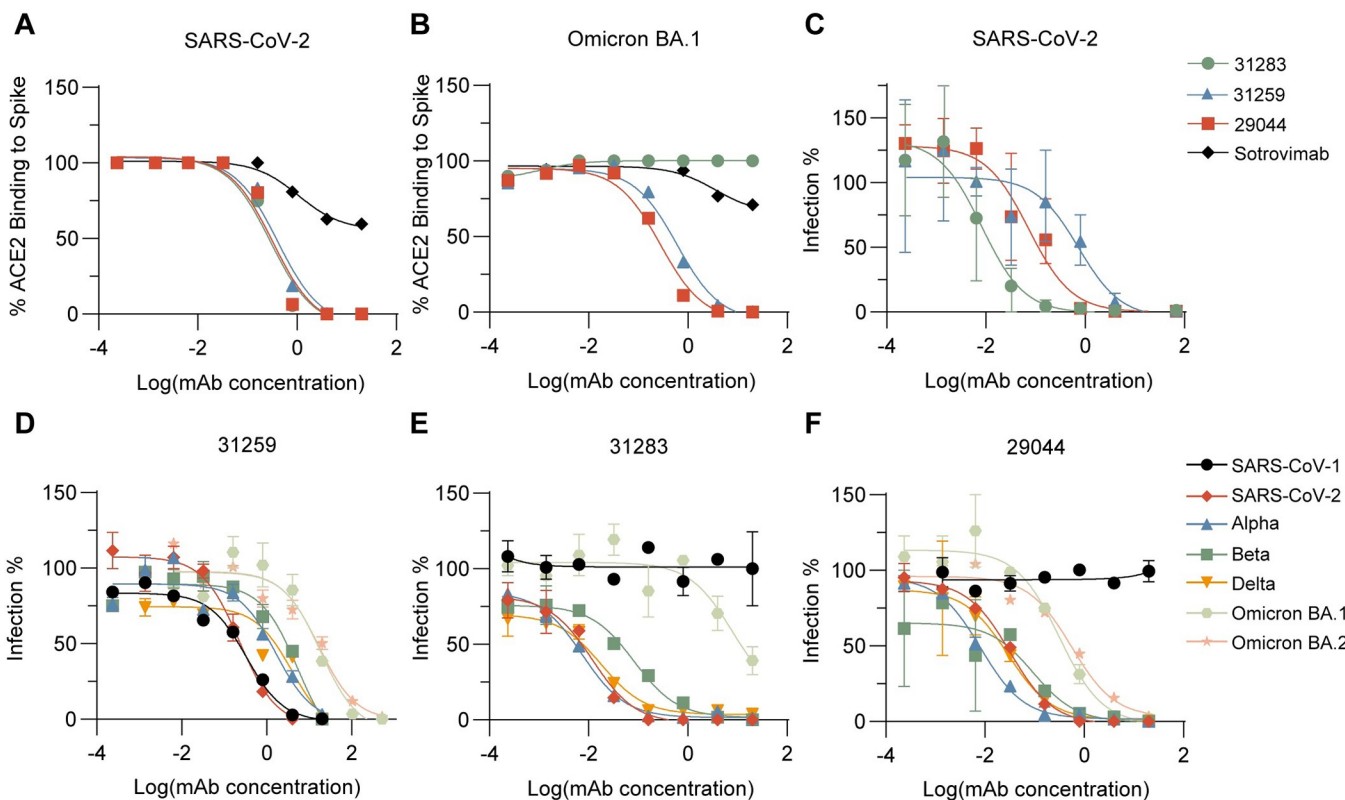

**Fig 6. Neutralization capacity and variant sensitivity of lead mAbs.** (A-B) ACE2-competition performed in duplicates at mAb 5-fold dilutions from 20μg/ml– 0.256ng/ml, towards SARS-CoV-2 Spike and Omicron BA.1 Spike proteins. Sotrovimab (black) shown as negative (non-ACE2-competitive) control. Three-parameter non-linear fit is plotted. (C) SARS-CoV-2 neutralization displayed as infection percentage at mAb 5-fold dilutions from 20μg/ml– 0.256ng/ml, towards full-length eGFP expressing SARS-CoV-2. Quantified as eGFP⁺ cells by flow cytometry. Error bars represent mean and range of triplicate determinations. Three-parameter non-linear fit is plotted. (D-F) Pseudovirus neutralization displayed as infection percentage at mAb 5-fold dilutions ranging from 517μg/ml– 0.256ng/ml, towards SARS-CoV-1 and selected SARS-CoV-2 variants of concern. Quantified as eGFP⁺ cells by flow cytometry. MAb identifier is shown as title on graph. Error bars represent mean and range of duplicate determinations. Three-parameter non-linear fit is plotted.

avert viral infection from all variants, while both 29044 and 31283 showed no neutralization effect towards SARS-CoV-1 in agreement with SPR data. Further, Omicron neutralization was present (albeit less potent compared to other stains) for both 31259 ($IC_{50}$ = 18.41 ug/ml) and 29044 ($IC_{50}$ = 0.32 ug/ml), while 31283 displays no neutralization of Omicron as predicted by ACE2 competition assays.

### In vivo prophylaxis

Finally, 31259, 31283 and 29044 were evaluated *in vivo* to determine their prophylactic potential. K18-hACE2 transgenic mice were used as a model of SARS-CoV-2 infection to study both viral titers in the lungs, and the disease course. The mAbs were administered 24 hours prior to SARS-CoV-2 exposure, and the mice were monitored daily for 4 days before harvest of lungs for analysis of viral titers (Fig 7A). We observed an up to three-fold dose dependent reduction in viral RNA copies/mg lung, for all three antibodies tested (S7A Fig). This dose dependent reduction was confirmed in SARS-CoV-2 outgrowth assays of lung tissues, showing up to 5-fold reduction in viral outgrowth, with multiple mice receiving high doses showing no discernable viral replication above limit of detection (S7B Fig). Overall, the mAbs with lower $IC_{50}$ values, 31283 and 29044, showed a larger reduction in viral copies/mg lung as well as replication competent virus in the lungs, compared to mAb 31259.

The antibodies were further evaluated in survival studies, where mice were monitored daily for up to 13 days after SARS-CoV-2 exposure, to determine if the observed reduction in virus observed in the lungs, would translate to disease protection. During these studies, pre-defined welfare criteria were applied to score the disease severity of the mice and determine time of euthanization. In Fig 7B, a clear difference was observed in the weight stability of the animals

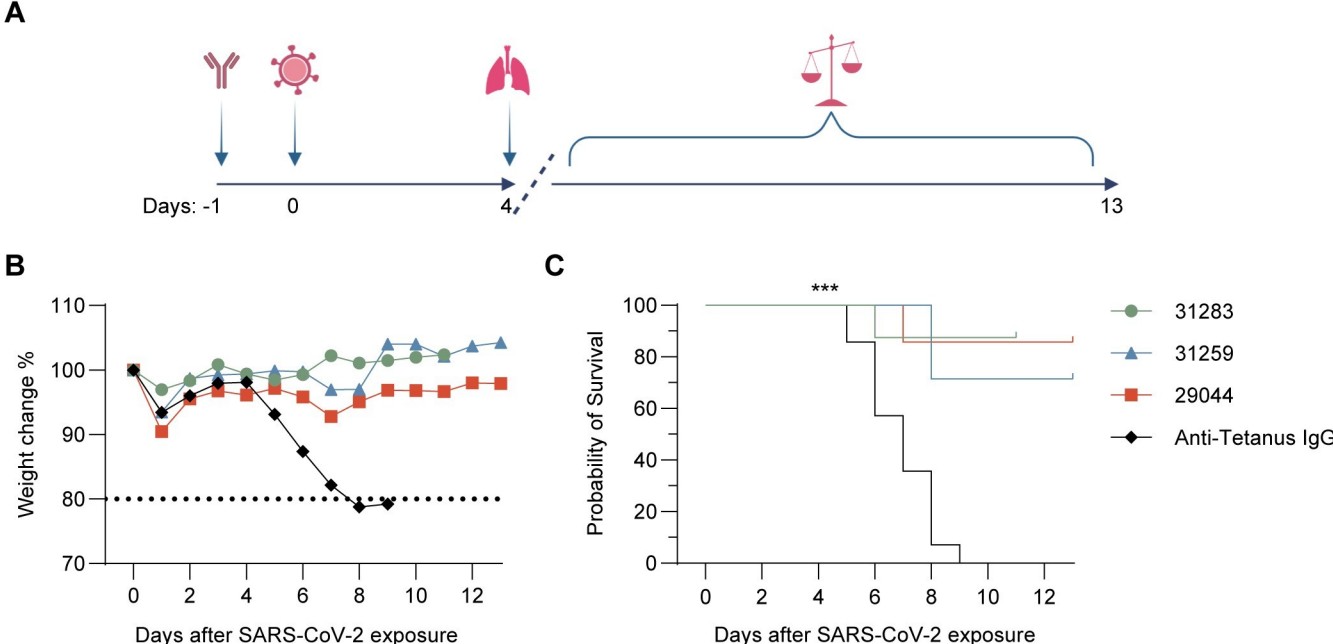

**Fig 7. In vivo prophylaxis.** (A) Schematic timeline of in vivo experiment design. Created with BioRender.com. (B) Daily percent weight change in mice post SARS-CoV-2 exposure. Day 0 indicates day of viral exposure. Anti-Tetanus IgG (black) was dosed 400 ug/mouse, n = 14. mAb 31283 (green) was dosed 400 ug/mouse, n = 8. mAb 29044 (red) was dosed 400 ug/mouse, n = 7. mAb 31259 (blue) was dosed 2 mg/mouse, n = 6. (C) Probability of survival displayed for the treatment groups shown in b. P-values calculated individually for each mAb treatment group compared to the Anti-Tetanus IgG control group using Log-rank (Mantel-Cox) test; 31283 p = 0.0002; 31259 p = 0.0004; 29044 p = 0.0003.

between the mAb treated groups compared to the rapid weight loss observed in control mice between days five and eight post SARS-CoV-2 exposure. This translated into a highly significant increased rate of survival within mAb treated mice compared to control mice (31283 p = 0.0002; 31259 p = 0.0004; 29044 p = 0.0003), as displayed in Fig 7C.

## Discussion

We aimed to isolate SARS-CoV-2 neutralizing mAbs from 15 COVID-19 convalescent participants, utilizing LIBRA-seq and single cell mRNA sequencing. 455 mAbs were selected from the participant's B cell population, for SARS-CoV-2 binding and neutralization screening. Overall, we found that while all mAbs were expressed and screened in IgG1 format, neutralizing mAbs that were originally expressed by the B cells as IgA1 (13 mAbs), IgA2 (2 mAbs) and IgG1 (18 mAbs) exhibited neutralizing activity. This indicated that the reformatting did not disrupt binding in all cases, although it cannot be ruled out that some IgA mAbs lost binding capacity. Of note, the two elite candidate mAbs 31283 and 31259 from LIBRA-seq both had IgG1 isotype, suggesting IgG1 should be the primary isotype selected in future studies. When combining neutralization screening and LIBRA-seq barcode data, Ag scores were not directly capable of highlighting cluster 8 as the best source of Ag-specific B cells. However, post hoc analysis did show that the Ag scores highly correlated with subsequent binding and neutralization capacity found in cluster 8. Previous studies using a similar method for Ag-specific B cell sorting have reported higher neutralization hit-rates within the mAb pools [23, 27, 28]. However, the molecules used for cell sorting and DNA oligo quantification in this study were dextramers, consisting of a dextran backbone carrying multiple acceptor sites for the binding of biotinylated molecules. Other similar LIBRA-seq studies have used direct DNA-oligo and fluorophore coupled Ags without the use of a dextran backbone, which may have contributed to a significant amount of background potentially due to avidity effects.

A commonly used classification system to describe SARS-CoV-2 neutralizing antibodies has been established by Barnes et al [26], distinguishing antibodies into 4 classes, based on binding to the up- or down-confirmation of the RBD, as well as within or outside the RBM. The reference mAbs Casivirimab and S2E12 are both class 1 antibodies, binding the RBM on open Spike formations [33, 34]. Class 1 antibodies are characterized by strong neutralizing potential, ACE2 competition, but with a general high sensitivity to point mutations in the RBM, evident from loss of effect against emerging VOC [26, 36, 37]. The presence of the most potent neutralizing antibodies reported in this work in epitope community 1, their overall loss of sensitivity to VOC in several binding assays, as well as the competition with Casivirimab and S2E12, collectively suggests these mAbs to be class 1 antibodies. Additionally, their abundance in the lead mAb pool is in line with studies identifying class 1 antibodies as the most commonly isolated class from convalescent patients [19, 25, 26]. Cryo-EM analysis of Omicron BA.1 - Fab 29044 epitope interaction, revealed binding of RBD in the up confirmation only, with 29044 binding interface overlapping the ACE2 footprint, conclusively assigning this mAb to class 1 up-RBD only [38]. Sotrovimab and Imdevimab are known class 3 antibodies, binding both up- and down-formations of the RBD [32–34, 37]. These antibodies bind outside of the ACE2 binding site (though ACE2 competition may still be seen due to steric interference), providing a generally higher variant binding breadth and mutation resistance, compared to RBM specific mAbs [26, 36]. While no reference antibodies were present in epitope community 2, the presence of ACE2 non-blocking mAbs and lack of competition towards class 3 antibodies, suggests that this community too binds an epitope outside the RBM, distinct to that of class 3 mAbs. The observed VOC binding breath of 31259, 31319 and 31414 mAbs by both mesoscale and SPR analysis, especially towards SARS-CoV-1, certainly suggests the targeting

of a distinct well-preserved epitope outside the RBM. These mAbs may provide an interesting foundation for future mAb engineering studies, pursuing development of more potent cross-neutralizing mAbs.

The static conformations of spike proteins during the various binding assays applied here may both over- and underestimate antibody binding of different VOC, risking the misinterpretation of important antibody-Spike interactions. The variant sensitivities observed from binding assays was functionally verified by pseudovirus neutralization assays for most of the elite neutralizer antibodies and particular the three mAbs; 31283, 29044 and 31259. These mAbs have distinct variant sensitivity profiles and originate from two different RBD binding epitope communities. Collectively, this advocates for the applied screening strategy as robust in identifying the antibodies of most potent neutralization and mutation resistant binding profiles.

The most time-consuming process of mAb discovery is by far the functional screening of large mAb libraries. The utilization of antigen specific B cell sorting in combination with single cell sequencing provides a distinct advantage compared to direct B cell receptor cloning. The collective data obtained for each individual B cell permits a qualified selection of the B cells more likely to produce mAbs of interest. As a result, the size of the mAb screening selection may be vastly reduced. This process may be further optimized in future searches, if the transcriptomic characteristics of SARS-CoV-2 neutralizing mAb producing B- ells are known as shown in this work. In general, memory B cells originate from germinal B cells of lower affinity maturation than plasma cells [39]. This has also been observed during SARS-CoV-2 infections [40]. We therefore expected to find the majority of neutralizing mAbs within the activated B cell clusters 6 and, showing high affinity maturation. Interestingly, this was only the case for cluster 8, with cluster 6 harboring very few neutralizing mAbs, with none of them potent enough to be selected as lead mAbs.

Previous studies have reported CD11c$^+$CD95$^+$ subsets of activated memory B cells, as observed here in cluster 6, as highly enriched SARS-CoV-2 antigen-labeled cells similarly identified by LIBRA-seq [23]. Cluster 6 is further characterized by expression of transcription factors and integrins associated with memory B cells prone to differentiation towards antibody secreting cells [31, 41, 42]. This may suggest cluster 6 as a population potentially originating from re-activated seasonal coronavirus memory cells, as suggested in previous studies [43], supporting their inferior neutralization potency of SARS-CoV-2. Similarly, the phenotype of activated memory B cells in cluster 6 shares several markers with memory B cells of extrafollicular (EF) origin known as double negative B cells (DN) that have been previously reported to be associated with critically ill SARS-CoV-2 patients [44, 45]. While the phenotypic conventions are still unclear [46], DN B cells are also known as atypical memory B cells, characterized by CD11c and T-bet expression, but lack of CD27 and IgD. While CD27 was detected in cluster 6, it was expressed at lower levels than in the large memory B cell population. DN B cells can be further subdivided into DN2 cells. Cluster 6 was associated with high expression of *FCRL5* and *FCRL3* that are both markers of DN2 cells. Therefore, cluster 6 potentially consist of DN2 B cells originating from an early EF response, possibly explaining the low binding and neutralization capacity.

In contrast, cluster 8 was characterized by high expression of actin and integrin molecules, the mobilization of which is known to play a crucial role during B cell activation, antigen presentation, T-cell interaction and indicative of migration to inflammatory sites upon activation [47–50]. Collectively, these interactions are suggestive of B cells undergoing somatic hypermutation, corroborated by the high hit-rate of potently neutralizing SARS-CoV-2 mAbs found within this cluster. Future mAb discovery efforts may hence be optimized through screening

mAbs from activated B cells with transcriptomic profiles similar to that identified in cluster 8, rather than activated B cells in general.

While antibody production is known to continue for months post initial SARS-CoV-2 infection, the majority of antibody producing plasma cells are found in lymphoid tissues rather than in circulation, at the median time of this cohorts inclusion [51, 52]. Hence, the time of blood sampling after viral exposure may be of great importance to the cells analyzed during mAb discovery, in contrast to the neutralization profiles patient selections are based on. Interestingly, recent studies have shown that SARS-CoV-2 antibody maturation may be observed up to a year after infection, displaying continued evolving towards a broader binding antibody pool [53, 54]. Looking at vaccine induced responses, initial antibodies are predominantly class 1 and 2. However, further vaccine boosters seem to result in an increase in class 3 antibodies with increased binding breadth [55, 56].

We isolated mAbs from convalescent patients infected at the origins of the SARS-CoV-2 pandemic. Since then, the emergence of viral mutational variants, therapeutic mAbs, and vaccinations have changed the landscape of the pandemic several times over. The continuous emergence of viral VOC escaping clinical mAbs has shown that broad binding mAbs with targets outside the RBM are advantageous compared to potent but mutation-vulnerable RBM binding mAbs. MAb 31259 discovered here may be a good example of a less potent, but broadly binding antibody with therapeutic capacity at an appropriate dose *in vivo*. The mAb 29044 displayed a lower $IC_{50}$ value, though lacking SARS-CoV-1 cross-reactivity, illustrating the possibly necessary trade-off between neutralization potency and viral variant binding breadth. However, the presence of such diverse mAbs in early pandemic cohorts shows great promise for the continuous possibility of discovering relevant, effective, therapeutic mAbs as the SARS-CoV-2 pandemic evolves. In conclusion, the LIBRAseq B cell transcriptomic cluster profiling, and functional mAb screening workflow presented here, may help facilitate future rapid SARS-CoV-2 mAb discoveries, aiding adaptation to the ongoing pandemic and future recurrent pathogens.

## Materials and methods

### Patient inclusion

A cohort of 203 SARS-CoV-2 convalescent individuals were included at the Department of Infectious Diseases at Aarhus University Hospital in spring 2020, and have been previously described by Vibholm *et al* [22] and Nielsen *et al*. [17]. Participant selection was based on the three following principles, with five patients selected from each: Group 1 had the highest plasma neutralization NT50 values, which was highly correlated to plasma levels of SARS-CoV-2 specific antibodies, assuming high levels of SARS-CoV-2 specific B cells would be present in these individuals. Group 2 the highest IC50 to SARS-CoV-2 specific antibody ratio, suggesting these participants may harbor more potently neutralizing antibodies. Group 3 the broadest profile of coronavirus targeting antibodies, including participants displaying SARS-CoV-1 and other more common coronavirus variant binding plasma antibodies. Informed written consent was given by all individuals prior to the study, which was approved by The National Health Ethics Committee (#1-10-72-76-20) and the Danish Data Protection Agency. Due to the pooling of PBMCs from five individuals none of the data pertaining to the monoclonal antibodies or B cell transcriptional data can be referred back to a specific individual.

### Recombinant SARS CoV-2 proteins

Four SARS-CoV-2 proteins were produced: RBD domain and the full trimeric spike proteins of SARS-CoV-2, the SARS-CoV-2 mutant D614G, as well as the RBD domain for the

SARS-CoV-1. The Ag designs were based on Uniprot P0DT2 sequence, with sequence AA14-1208 coding for SARS-CoV-2 (Wuhan strain) spike protein and P59594 for SARS-CoV-1. RBD domains encoding AA319-541 were made from both SARS-CoV-1 and SARS-CoV-2 spike proteins. Ag constructs were designed with an Avitag added C-terminal to allow site-specific biotinylation and an octa-HIS tag was inserted C-terminal to allow Ni-NTA purification. All constructs were transiently expressed in 200–500 mL Expi293F cells (ThermoFisher, Cat: A14635, Mycoplasma free) and purified according to an optimized Ni-NTA protocol. Purified proteins were analyzed by SDS-PAGE, SEC-MALS and tested for ACE2 binding by Biolayer Interferometry (BLI). The proteins were site-specific biotinylated using BirA enzyme and validated by SDS-PAGE gel-shift protocol [57]. Antigen sequences can be found in S1 Fig.

## Antigen specific FACS sorting of memory B cells and single cell sequencing

PBMCs from convalescent Covid-19 patients were pre-enriched for B-cells using EasySep human Pan-B-cell enrichment kit (19554, Stemcell Technologies, Cambridge, UK). B-cells were sorted by FACS on an Aria III cell sorter by sorting antigen specific memory B cells using a pool of 4 antigens as described in the Recombinant SARS CoV-2 proteins paragraph. The antigens were bound to dextramers (dCODE Klickmer, Immudex, Virum, Denmark) barcoded with a DNA oligo and coupled with both PE fluorochrome and Streptavidin. Each of the 4 antigens were mixed separately with a different DNA oligo barcode biotin-dextramer. A mix of human surface markers CD19, IgG and IgA and a live/dead cell marker were mixed with cells and 4 klickmer/antigens. Cells expressing CD19, IgG and IgA and binding to dextramer encoded antigens were sorted into bulk for 10x Genomics workflow and used for single cell sequencing with Chromium Next GEM Single Cell 5' Kit v2 (10x Genomics, CA, USA). Gating strategy is shown in S2 Fig.

## Data processing of raw 10x Genomics data

All data processing was carried out on the High-Performance Computing cluster Computerome 2.0. Raw fastq data were processed with cellranger vs. 6.0 multi pipeline, generating UMI count matrices for all detected cells as well as VDJ-specific analyses. The raw count matrices for each sample were loaded into R where the following data processing was carried out using the Seurat 4.0 package [58]. The samples were subject to standard QC based on the 1) percent mitochondrial genes detected (>7.5% filtered), 2) the total number of unique features (<200 features filtered) and 3) total number of feature counts (>4000 counts filtered). Expression levels were log-normalized using the NormalizeData() function. The 4000 most variable features were identified using FindVariableFeatures(). The following gene families were excluded from the variable features: *IGKV*, *IGLV*, *IGHV*, *IGHJ*, *IGHD* and *AC233755*. CellCycleScoring() was applied to remove any confounding effect from cell cycle state. Data were then scaled such that each feature has mean zero and unit variance using ScaleData() while regressing out the contribution from mitochondrial gene expression and cell cycle. Scaled data was subjected to principal component analysis with PCA(). Utilizing Elbow and Jackstraw plots, 40 principal components were identified to capture most of the variance in the data. Harmony-based batch correction was subsequently applied to remove any effects caused by technical variance [59], also defining 40 harmony embeddings. The cells were clustered with FindNeighbors() with the 40 harmony embeddings, and FindClusters() with 0.5 resolution. A two-dimensional embedding using UMAP was calculated with RunUMAP() for visualizing the cell populations.

The final set of cells was the basis for differential gene expression analysis with FindAllMarkers() with a minimum of cells expressing the marker at 25% and a log fold change threshold of 0.25. Differential gene expression analysis revealed a cluster defined by the NK markers

(*NKG7*, granulysin (*GNLY*), granzyme B (*GZMB*), perforin genes (*PRF1*) and *CD3*). These cells were excluded, and the workflow outlined above repeated. Expression levels of specific markers genes were visualized with violin plots and heatmaps with VlnPlot() and FeaturePlot (). The UMAP coordinates and clustering were exported from R and mapped to the other data based on unique cell barcode.

### Repertoire sequence analysis

All obtained VH and VL antibody sequences were analyzed using CLC Main Workbench (Qiagen software package). This workflow identified serious sequence liabilities, also referred to as developability issues. These issues included reading frame issues, stop codons, unpaired cysteines in CDRs, *IGHV4-34* germline [60], N-linked glycosylation motifs and sequences with extreme predicted hydrophobicity based on the GRAVY score index when compared to 242 antibodies in clinical development. Impact of deselecting these liabilities is shown in S4 and S5 Figs. Sequence clustering was based on the clonotype predicted with partis [61], which predicting the most likely naïve sequence origin of each mAb (the V(D)J recombination event). This was applied as the basis for calculating the somatic hypermutation rate. The antigen score (from feature barcodes) for each cell was calculated by log2p1 normalization of the number of features barcode-associated UMIs per cell, divided by the total number of UMIs for the cell, multiplied by 10,000. This is analog to the normalization performed for the GEX data but using log2 instead of the natural logarithm.

### Synthesis and expression of identified mAbs

Amino acid sequences for VH and VL from the VDJ data for selected cells were reverse translated and codon optimized to expression in ExpiCHO cells (ThermoFisher, Cat:A29127, Mycoplasma tested) using the TWIST web interface (TWIST Bioscience (CA, USA). The resulting DNA sequences were synthesized and, for VH, cloned into IgG1 wild-type pcDNA 3.4 vectors, whereas VL DNA sequences were cloned into kappa- or lambda vectors depending on the light chain locus expressed by the B cell. Full-length IgG1 antibodies were expressed by transfection of the cognate heavy and light chain plasmids into ExpiCho-S cells as described by Meijer *et al* [62]. Supernatants containing full-length IgG1 antibodies were harvested after 11 days and used to screen for Ag binding. A subset of mAb supernatants were purified using standard protein A column chromatography (Protein A PhyTip columns) and buffer exchanged into PBS.

### Binding kinetics measured by surface plasmon resonance

Binding kinetics to antigen were measured by SPR, Carterra LSA. Binding to the following antigens was characterized; The trimeric complex and RBD of SARS-CoV-2, and the RBD of SARS-CoV-1 were produced as described above. The SARS-CoV-2 (COVID-19) RBD (N501Y), SARS-CoV-2 Spike RBD (K417N, L452R, T478K), SARS-CoV-2 Spike RBD (L452R, E484Q), and SARS-CoV-2 Spike RBD (B.1.1.529/Omicron) were from ACRO Biosystems (Newark, Delaware US).

To prepare a capture surface, Goat-anti-human IgG Fc (Southern Biotech) was amine-coupled onto a HC200M (Carterra) sensor by the Single Flow Cell (SFC). The surface was activated by 0.4 M EDC, 0.1 M sulfo-NHS, and 0.1 M MES, pH 5.5 for 5 min, followed by immobilization of 75 μg/mL Goat-anti-human IgG Fc diluted in 10 mM sodium acetate, pH 4.5 for 10 min. Last, excess reactive esters were quenched for 3 min by injection of 1 M ethanolamine, pH 8.5, for 7 min. The instrument was primed in running buffer (PBS pH 7.4, 0.05% Tween-20). Antibodies were captured as individual spots on the chip for 12 minutes by

the 96 print head (96PH) as triplicates or more of each antibody. Kinetic analysis was performed by applying kinetic titration series of antigen at increasing concentrations. Association and dissociation of antibodies were recorded for 10 minutes each. After end cycle, the surface was regenerated by 0.45% $H_3PO_4$ for 2×20 s and washed for 5 min in running buffer. Binding responses were local ROI referenced, buffer blanked and cropped using Carterra's KIT software tool. Processed data were fitted to a simple Langmuir 1:1 binding model for calculation of the on-rate ($k_{on}$ or $k_a$), off-rate ($k_{off}$ or $k_d$) and affinity ($K_D$) constants of each antibody.

## Epitope binning

Paired antibody competition was performed by SPR, IBIS MX96. Anti-SARS-CoV-2 antibodies were diluted to 15 µg/ml in PBS and spotted onto a G-a-hu-IgG Fc SensEye by capturing for 15 minutes using a Continuous Flow Microspotter (CFM). After sensor preparation, the sensor was docked in the SPR imager and residual binding sites were blocked by Herceptin (trastuzumab) followed by chemical cross-linking by SensEye FixIt kit (IBIS, Netherlands). Antibody competition analysis was performed using a Classical Sandwich Assay setup. The SARS-CoV-2 RBD antigen was diluted to 50 nM in PBS, 0.05% tween20, 200 nM Herceptin running buffer and injected for 5 min followed by 5 min injection of the 100 nM antibody to establish antibody competition patterns. Between each injection the surface was regenerated for 30 s by 100 mM $H_3PO_4$, pH 3. Recombinant ACE2 (Acro Biosystems) was included in the assay as 50 nM injection to measure antibody blocking of the receptor. Data was analyzed by Epitope Binning 2.0 (Wasatch, USA) applying user-defined threshold settings to classify antibodies as either blocked or not blocked (sandwiching) analyte-ligand pairs.

## Cryo-EM sample preparation, data collection and image processing

Equal volumes of 0.3 mg/ml of SARS-CoV-2 spike trimer (B.1.1.529/Omicron) and 0.3 mg/ml of Fab 29044 were incubated at room temperature for 5 minutes. 3 µL of the solution containing the complex was then loaded onto the plasma cleaned Protochip AuFlat Grids 0.6/1 grids. The grids were blotted using a Vitrobot Mark IV (Thermo Fisher) with the following settings: 4˚C, 100% humidity, no wait time, 3–5 sec blot time, blot force 10 and plunge frozen in liquid ethane pre-cooled by liquid nitrogen. The grids were analyzed by recording micrographs using a Titan Halo (Thermo Fisher) electron microscope operating at 300 kV and equipped with a Gatan K3 camera. Around 5000 movies were collected at a nominal magnification of 37,000x with the calibrated pixel size of 0.8465 Å/pixel and total electron exposure of ~ 75 e-/$Å^2$ fractioned into 80 frames using SerialEM [63].

Motion correction, CTF estimation and particle extraction using a box size of 420 pixels were done in Warp [64] which resulted in a particle stack containing 909,583 particles. Further processing including 2D and 3D classification and refinements and 3D reconstruction of cryo-EM maps were done in cryoSPARC [65]. Initial particle cleanup was done using a round of 2D classification and Select 2D classes yielding 351,688 particles. These particles were subjected to ab initio 3D reconstruction to generate initial 3D maps with apparent architecture of Spike trimer-Fab complex. Further round of particle cleanup was done using heterogenous refinement with ab initio spike trimer-fab and decoy maps as input volumes resulting in a final set of 290,076 particles. These particles were subjected to 3D classification using heterogenous refinement. The resulting 3D classes were refined using non-uniform refinement [66] with C1 symmetry to improve the resolution. One of the non-uniform refined 3D classes with 103,699 particles showed density of Fab 29044 bound to SARS-CoV-2 spike trimer (B.1.1.529/Omicron) and was used for model building and analysis. The estimated resolution for this class is 3.82 Å based on gold-standard Fourier shell correlation of 0.143 criterion.

## Cryo-EM structure modeling, refinement and analysis

A homology model of the variable region (Fv) of Fab 29044 was built using RosettaAntibody protocol implemented in ROSIE server [67]. Initial coordinates for Omicron spike trimer were taken from PDB ID 7WLY. Initial models of Fab 29044 Fv and Omicron spike trimer were first rigid-body fitted into the cryo-EM density map using CHIMERA [68]. The rigid-body fitted antigen-antibody complex was then subjected to molecular dynamics flexible fitting using Namdinator [69] followed by iterative rounds of real space refinement and model building in PHENIX [70] and COOT [71]. Analysis of structures and generation of structural graphics were done with ChimeraX 1.4 [72], PyMOL [73] and PDBePISA [35].

## Mesoscale binding assay

All serum samples and expressed mAbs were screened for binding to various coronavirus proteins as previously described [17]. Briefly, antibody levels were measured using the MSD Coronavirus Plate 1 (Spike epitopes) and MSD Coronavirus Plate 9 (viral variants) (Meso Scale Discovery, Rockville, Maryland), following the manufacturers guidelines. Unspecific antibody binding was blocked using MSD Blocker A. Patient plasma was diluted 1:4630 and mAbs were diluted to 2ug/ml in Diluent100, before application to the plates. All antibodies were screened as supernatants. After sample incubation, bound IgG was detected by incubation with MSD SULFO-TAG Anti-Human IgG Antibody and subsequently measured on a MESO QuickPlex SQ 120 Reader after addition of GOLD Read Buffer B.

## Mesoscale ACE2 competition assay

All serum samples and expressed mAbs were screened for their ability to compete with ACE2 interaction to various coronavirus proteins as previously described [17]. Briefly, Spike and RBD targeting antibodies with the ability to compete with ACE2 binding were measured using the MSD Coronavirus Plate 1 or 9 as above. Patient plasma was diluted 1:10 and purified mAbs were diluted to 2ug/ml in Diluent100. COVID-19 blocking antibody calibrator and diluted samples were incubated after plate blocking. SULFO-Tag conjugated ACE2 was added before washing, allowing ACE2 to compete with antibody binding to spike and RBD antigens immobilized on the plate. Bound ACE2 was detected as described for the serology assay above, and antibody concentrations were subsequently calculated using the MSD Discovery Work-bench software. For lead mAbs, binding and ACE2 competition sensitivity to SARS-CoV-2 spike receptor binding domains (RBD) containing point mutations from variants of concern was also analyzed using the MSD Coronavirus Plate 9. Further, MSD Coronavirus Plate 23 containing spike from the omicron variant was used to assess mAb dilutions 20ug/ml–0.256ng/ml, and IC50 values were calculated.

## Neutralization assays

Pseudovirus production and neutralization assays were performed as described previously [17]. Briefly, either pseudovirus expressing coronavirus spike proteins variants of interest (B.1.1.7, B.1.351, P.1, B.1.617.2, BA.1, BA.5, SARS-CoV-1) (Elbe and Buckland-Merrett *et al*, 2017) [74], or full-length SARS-CoV-2 engineered to express eGFP [75], were applied for analysis. Plasma samples were heat inactivated at 56˚C for 45min prior to analysis; Eight 5-fold dilutions in DMEM with 10%FBS and 50U/mL P/S (cDMDM) of plasma was mixed with virus at MOI = 0.01 and incubated at 37˚C for 1 hour before addition of Vero TRMPSS2 SARS-CoV-2 [15] (Mycoplasma tested) permissive cells and further incubated for 24 hours at 37˚C. Final total plasma dilutions rested ranged from 1:25–1:1953125. All antibodies were initially

screened as supernatants at single concentration at 20ug/ml and 0.8ug/ml, after which full neutralization curves were performed on purified lead antibody candidates, in 5-fold dilutions in duplicates as described for plasma samples. The final total concentrations tested ranged from 517ug/ml– 0.256ng/ml. Cells were fixed in 1% PFA for at least 15 min at 4˚C, prior to analysis of cell eGFP expression on a Miltenyi Biotec MACSquant16 flow cytometry.

## In vivo studies

Animal experiments were conducted under animal license 2020-15-0201-00726 approved by the Danish Animal Experiment Inspectorate. Heterozygous B6.Cg-Tg(K18-ACE2)2Primn/J mice (JAX, Strain no:034860) at least 8 weeks of age were used for *in vivo* studies. Both males and females were included, the genders balanced between treatment groups. The mice were fed standard chow, and housed in ventilated Scanbur cages with 12 hour night/day light cycles. Experiments were performed in biological agent class 3 approved laboratories. mAbs 31283, 29044 and 31259 were administered I.P. to the animals 18–24 hours prior to exposure to coronavirus. A human IgG anti-tetanus antibody was administered to the control groups. 4ug, 40ug, 400ug, 700ug or 2mg mAb (700ug and 2 mg only for mAb 31259) total mAb was dosed for initial dose finding studies, with 400ug total 31283 and 29044 mAb and 2 mg 31259mAb dosed to each animal for the survival study.

The day after antibody treatment, animals were sedated with 75ug/g Ketamin and 1ug/g Medetomidin and exposed to 11.8 $TCID_{50}$ SARS-CoV-2 (SARS-CoV-2/München-1.1/2020/929) [75] through the nostrils. Post exposure, 1ug/g Atipamezol was administered prior to the mice being returned to their home cages to wake up.

The animal's weight was monitored daily after viral exposure. For survival studies, all mice were monitored daily until 20% weight loss occurred, or signs of inactivity and/or labored breathing was observed, after which they were euthanized. This observation was carried out by animal technicians, blinded to the treatment groups.

For dose treatment studies, the mice were euthanized by cervical dislocation on day 4 post viral exposure, and the lungs were harvested for viral quantification and outgrowth. Immediately after removal, both lungs were weighted, and homogenized in RNAse free PBS using a TissueLyser LT (Qiagen). The tubes were briefly spun down, and 2/3 of the supernatant was frozen directly on dry ice. Lysis buffer (Qiagen RNeasy mini kit) was subsequently added to the remainder of the lung tissue and supernatant. The sample was homogenized again for 3min at max speed on the TissueLyser, and frozen immediately on dry ice.

## ddPCR

RNA was purified from 200ul of the lung samples homogenized with lysis buffer, using the RNeasy mini kit (Qiagen) following the manufacturer's instructions. 22 µL ddPCR reaction mixes were prepared in duplicates according to the Bio-Rad protocol, containing 5.5 µL One-Step RT-ddPCR Advanced Kit for Probes (Cat #:1864022), 2.2 ul Reverse Transcriptase, 1.1 ul 300 mM DTT, 250 nM nCOV_N1 Probe (IDT Cat #10006823), 1000 nM of each forward and reverse nCOV_N1 primers (IDT Cat#: 10006821/10006822), and 10 µL RNA sample, diluted 1 million times. Droplets were generated using the QX200 Droplet generator (Bio-Rad) and amplified in a C1000 Touch Thermal Cycler (Bio-Rad) under the following conditions; 25˚C for 3 min, 50˚C for 1 h, 40 cycles of 95˚C for 30 s and 55˚C for 1 min, lastly 98˚C for 10 min and infinite hold on 12˚C. Droplets were subsequently analyzed by a QX200 droplet reader (Bio-Rad) and data analysis was performed in QuantaSoftTM analysis software (Bio-Rad). Total concentration per sample was calculated based on total N1 concentration per reaction, and normalized to lung volumes for the mice individually.

## TCID50

An estimation of infectious viral units in lung homogenates was carried out for each mouse. TCID50 analysis was set up as follows: Five-fold serial dilutions of each mouse lung homogenate was made in DMEM containing 10% FBS and 50 U/mL P/S. 50 μL of each dilution was incubated with 2,000 Vero76 cmyc hTMPRSS2 cells, in 50 μL DMEM containing 10% FBS and 50 U/mL P/S. Plates were incubated at 37˚C for 72 hours. The final total dilutions of lung homogenates ranged from 1:50–1:97656250 and were analyzed in eight replicates. Cytopathic effect (CPE) was determined manually as present or absent in each well, using a light microscope, and used to calculate $TCID_{50}$/ lung volume for the mice individually.

## Data and statistical analysis

Flow cytometry data was analysed using FlowJo (Version 10.7.1). All mAb screening data *in vitro* and *in vivo* was processed and graphed in GraphPad Prism version 9.2.0. Neutralization curves were plotted with three parameter non-linear fits, from which $IC_{50}$ and $NT_{50}$ values were calculated. Mice were randomized into treatment groups, balancing genders and age. Group size was determined using power calculations. Values in heat maps displaying binding, were normalized within each column. Heat map fold-changes were calculated between log-transformed values, normalized between 0 and 100 based on the highest and lowest value for each individual viral variant displayed. Heat map ACE-competition percentages were calculated from inter-assay calibration curves. P-values are indicated as follows: n.s. = not significant,* = $p \leq 0.05$, ** = $p < 0.01$, *** = $p < 0.001$, **** = $p < 0.0001$.

## Supporting information

**S1 Fig. Protein antigens.**
(PDF)

**S2 Fig. FACS gating strategy.**
(PDF)

**S3 Fig. Characteristics of the single-cell population.**
(PDF)

**S4 Fig. Drug development liabilities.**
(PDF)

**S5 Fig. Cluster distribution of drug development liabilities.**
(PDF)

**S6 Fig. Surface plasmon resonance competition data.**
(PDF)

**S7 Fig. SARS-CoV-2 lung viral loads.**
(PDF)

**S8 Fig. Impact of germline on selection/deselection.**
(PDF)

**S9 Fig. Surface plasmon resonance sensorgrams of RBD variants.**
(PDF)

**S10 Fig. Cryo-EM refinement.**
(PDF)

**S1 Table. Surface plasmon resonance data.**
(PDF)

**S2 Table. Data on top neutralizing monoclonal antibodies.**
(PDF)

**S3 Table. Binding kinetic values of mAb.**
(PDF)

**S4 Table. Parameters and statistics for cryo-EM data collection, processing, structure refinement, and validation.**
(PDF)

## Acknowledgments

We would like to thank all the individuals in the study for the kind donation of both their time and biological material.

## Author Contributions

**Conceptualization:** Klaus Koefoed, Martin Tolstrup.

**Data curation:** Randi Westh Hansen, Harsh Bansia, Mike Vestergaard, Frederik Holm Rothemejer, Anne Worsaae, Deijona Brown, Maria Lange Pedersen, Anna Karina Juhl, Mariane Høgsbjerg Schleimann.

**Formal analysis:** Stine Sofie Frank Lende, Nanna Møller Barnkob, Randi Westh Hansen, Harsh Bansia, Maria Lange Pedersen, Anna Halling Folkmar Rahimic, Amédée Des Georges, Mariane Høgsbjerg Schleimann, Klaus Koefoed, Martin Tolstrup.

**Funding acquisition:** Klaus Koefoed, Martin Tolstrup.

**Investigation:** Stine Sofie Frank Lende, Mariane Høgsbjerg Schleimann, Martin Tolstrup.

**Methodology:** Stine Sofie Frank Lende, Nanna Møller Barnkob, Mike Vestergaard, Maria Lange Pedersen, Anna Halling Folkmar Rahimic, Anna Karina Juhl, Mariane Høgsbjerg Schleimann, Martin Tolstrup.

**Project administration:** Martin Tolstrup.

**Resources:** Laurent-Michel Vuillard, Martin Tolstrup.

**Supervision:** Torben Gjetting, Lars Østergaard, Amédée Des Georges, Mariane Høgsbjerg Schleimann, Klaus Koefoed, Martin Tolstrup.

**Writing – original draft:** Stine Sofie Frank Lende, Nanna Møller Barnkob, Klaus Koefoed, Martin Tolstrup.

**Writing – review & editing:** Stine Sofie Frank Lende, Nanna Møller Barnkob, Martin Tolstrup.

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
