## [Decision Letter · Decision Letter 0]

29 May 2023

PONE-D-23-12939Discovery of neutralizing SARS-CoV-2 antibodies highly enriched in a unique antigen specific B cell clusterPLOS ONE

Dear Dr. Tolstrup,

Thank you for submitting your manuscript to PLOS ONE. After careful consideration, we feel that it has merit but does not fully meet PLOS ONE’s publication criteria as it currently stands. Therefore, we invite you to submit a revised version of the manuscript that addresses the points raised during the review process.

We look forward to receiving your revised manuscript.

Kind regards,

Nagarajan Raju

Academic Editor

PLOS ONE

Journal Requirements:

   "We would like to thank all the individuals in the study for the kind donation of both their time and biological material. This study was supported by a grant from the Danish Ministry for Research and Education (grant# 0238-00001B) and The Danish Innovation Fund (grant# 0208-00018B). SFL was supported by a scholarship from Aarhus University."

   "This study was supported by a grant from the Danish Ministry for Research and Education (grant# 0238-00001B to MT) and The Danish Innovation Fund (grant# 0208-00018B to MT and KK). SFL was supported by a scholarship from Aarhus University. The funders had no role in study design, data collection and analysis, decision to publish, or preparation of the manuscript."

Additional Editor Comments:

I suggest the authors to go through all the comments from reviewers and address them in the revised version of the manuscript

Reviewers' comments:

Reviewer's Responses to Questions

**Comments to the Author**

1. Is the manuscript technically sound, and do the data support the conclusions?

Reviewer #1: Yes

Reviewer #2: Yes

2. Has the statistical analysis been performed appropriately and rigorously? 

Reviewer #1: Yes

Reviewer #2: Yes

3. Have the authors made all data underlying the findings in their manuscript fully available?

Reviewer #1: Yes

Reviewer #2: Yes

4. Is the manuscript presented in an intelligible fashion and written in standard English?

Reviewer #1: No

Reviewer #2: Yes

5. Review Comments to the Author

Reviewer #1: The study by the Tolstrup lab describes the discovery and characterizati on of neutralizing SARS-CoV-2

anti bodies . This is a very complete study with a clear and comprehendible experimental design

protocol, convincing and complete data and conclusions supported by well controlled study. The

manuscript can be published as submitt ed

Reviewer #2: The manuscript by Lende et al is comprehensive and detailed, with good observations and analysis of data obtained using several sophisticated techniques. The approach was to classify B cells from convalescent patients into transcriptomic profile groups. Single-cell sequence information was used to generate monoclonal antibodies via cloning, transfection and expression in ExpiCho-S cells. Resulting monoclonal antibodies were then screened for antigen binding, neutralisation activity against SARS-CoV-2 pseudoviruses and authentic viruses including major variants of concern. Finally, highly neutralising or broadly neutralising monoclonal antibodies were tested for in vivo protection against disease in a mouse model, validating the predictive value of the transcriptomic profiling approach.

The manuscript is generally well written, in clear standard English but will need thorough proof-reading to correct e.g. spacing errors.

Requested clarifications:

In the abstract and throughout the main body of the text, the step from characterising B cells to screening mAbs is skipped over. Only in the methods section is the generation of mAbs from the sequence data even mentioned. It would greatly assist readers’ understanding the concept of the approach used if the mAb generation was made clearer when describing the study.

Detailed characterisation was carried out on antibody 29044, but this was ‘identified in a pilot run’ (line 150). Were transcriptomic profiling and screening carried out on this pilot run? If they were, can the information be compiled into that for the main run? If they were not, then including this antibody as an exemplar of the profiling approach as an effective pre-screening method is inappropriate and misleading.

Have the authors considered population or demographic differences between studies as potentially contributing to the differences identified in lines 91-94?

Why were switched isotypes IgG2, IgG3 and IgG4 excluded (line 125), as these were highly mutated from the germline?

Line 154: “The vast majority of neutralizing mAbs were RBD binders” however Figure 4A shows that most neutralising mAbs have weaker binding to the RBD than to the overall Spike.

Please include results from the in vivo dose finding studies.

Line 316: So what timepoint of blood sampling is recommended for mAb discovery? Was there any attempt to correlate neutralising antibody identification with time of PBMC sampling? Table 1 describes the time from diagnosis to inclusion ie PBMC sampling, but would time from symptom onset be a better indicator for participant groups 2 & 3?

Minor comments

Line 95: needs to refer to Fig 2E as well as 2D.

Line 111: CD80 is not shown in Fig 2

Lines 162-164: add comment on BA.1

Line 172: There is no blue in Fig 5A. The legend and Fig 5A describe Groups A-C, and text describes Groups 1-4. Please make text consistent with figure.

Line 205: Gamma is not included in Fig 6D-F.

Line 206: Fig 6D-F shows BA.2, not BA.5

Line 314: “mAb” in both instances on this line is probably actually referring to “antibodies”, not monoclonals.

Resolution & image quality of figures, especially Fig 2, could be better.

6. PLOS authors have the option to publish the peer review history of their article (what does this mean?). If published, this will include your full peer review and any attached files.

Reviewer #1: No

Reviewer #2: No

---

## [Author Response · Author response to Decision Letter 0]

7 Aug 2023

We have included a response to reviewer comments addressing all comments raised by the reviewers

---

## [Decision Letter · Decision Letter 1]

23 Aug 2023

Discovery of neutralizing SARS-CoV-2 antibodies highly enriched in a unique antigen specific B cell cluster

PONE-D-23-12939R1

Dear Dr. Tolstrup,

We’re pleased to inform you that your manuscript has been judged scientifically suitable for publication and will be formally accepted for publication once it meets all outstanding technical requirements.

Kind regards,

Nagarajan Raju

Academic Editor

PLOS ONE

Additional Editor Comments (optional):

Reviewers' comments:

Reviewer's Responses to Questions

**Comments to the Author**

1. If the authors have adequately addressed your comments raised in a previous round of review and you feel that this manuscript is now acceptable for publication, you may indicate that here to bypass the “Comments to the Author” section, enter your conflict of interest statement in the “Confidential to Editor” section, and submit your "Accept" recommendation.

Reviewer #2: All comments have been addressed

2. Is the manuscript technically sound, and do the data support the conclusions?

Reviewer #2: (No Response)

3. Has the statistical analysis been performed appropriately and rigorously? 

Reviewer #2: (No Response)

4. Have the authors made all data underlying the findings in their manuscript fully available?

Reviewer #2: (No Response)

5. Is the manuscript presented in an intelligible fashion and written in standard English?

Reviewer #2: (No Response)

6. Review Comments to the Author

Reviewer #2: (No Response)

7. PLOS authors have the option to publish the peer review history of their article (what does this mean?). If published, this will include your full peer review and any attached files.

Reviewer #2: No

---

## [Editor Report · Acceptance letter]

10 Sep 2023

PONE-D-23-12939R1 

Discovery of neutralizing SARS-CoV-2 antibodies enriched in a unique antigen specific B cell cluster 

Dear Dr. Tolstrup:

I'm pleased to inform you that your manuscript has been deemed suitable for publication in PLOS ONE. Congratulations! Your manuscript is now with our production department. 

Kind regards, 

on behalf of

Dr. Nagarajan Raju 

Academic Editor

PLOS ONE